# Quantitative Study on Solubility Parameters and Related Thermodynamic Parameters of PVA with Different Alcoholysis Degrees

**DOI:** 10.3390/polym13213778

**Published:** 2021-10-31

**Authors:** Siqi Chen, Hao Yang, Kui Huang, Xiaolong Ge, Hanpeng Yao, Junxiang Tang, Junxue Ren, Shixue Ren, Yanli Ma

**Affiliations:** 1Key Laboratory of Bio-Based Material Science and Technology of Ministry of Education, College of Materials Science and Engineering, Northeast Forestry University, Harbin 150040, China; csq18738668529@163.com (S.C.); yanlima@nefu.edu.cn (Y.M.); 2College of Material Science, Engineering Northeast Forestry University, Harbin 150040, China; yanghao1703@163.com (H.Y.); m15072122131@163.com (K.H.); gxl15665686206@163.com (X.G.); 18609377154@163.com (H.Y.); tjx0527@163.com (J.T.); 3School of Astronautics, Beihang University, Beijing 100191, China; rjx_buaa@163.com

**Keywords:** PVA, solubility parameter, IGC, alcoholicity, molecular dynamics simulation, surface properties

## Abstract

In recent years, inverse gas chromatography (IGC) and molecular dynamics simulation methods have been used to characterize the solubility parameters and surface parameters of polymers, which can provide quantitative reference for the further study of the surface and interface compatibility of polymer components in the future. In this paper, the solubility parameters and surface parameters of two kinds of common alcoholysis, PVA_88_ and PVA_99_, are studied by using the IGC method. The accuracy of the solubility parameters obtained by the IGC experiment is verified by molecular dynamics simulation. On the basis of this, the influence of repeated units of polyvinyl alcohol (PVA) on solubility parameters is studied, so as to determine the appropriate chain length of the PVA for simulation verification calculation. The results show that the solubility parameters are not much different when the PVA chain length is 30 and above; the numerical trends of the solubility parameters of PVA_88_ and PVA_99_ at room temperature are the same as the results of molecular dynamics simulation; the dispersive surface energy $\gamma_{s}^{d}$ and the specific surface energy $\gamma_{s}^{\mathrm{sp}}$ are scattered with the temperature distribution and have a small dependence on temperature. On the whole, the surface energy of PVA_99_ with a higher alcoholysis degree is higher than that of PVA_88_ with a lower alcoholysis degree. The surface specific adsorption free energy ($\Delta G^{\mathrm{sp}}$) indicates that both PVA_88_ and PVA_99_ are amphoteric meta-acid materials, and the acidity of PVA_99_ is stronger.

## 1. Introduction

Polyvinyl alcohol (PVA) is a kind of environmental friendly water-soluble polymer material [1] with good film-forming property, excellent performance, non-toxicity, and degradability [2,3,4]. It is a polymer obtained by the polymerization of vinyl acetate and hydrolysis of polyvinyl acetate [5,6], and it is widely used in the fields of food packaging, pharmaceutical film [7], biological tissue, and other composite materials [8,9,10,11]. As the application of PVA becomes more and more extensive, its good performance makes it a hot spot in the field of materials research. 

The properties of PVA are mainly affected by its average polymerization degree and alcoholysis degree [12,13,14,15]. The properties of PVA are different because of the different degree of polymerization and hydrolysis (alcoholysis) of PVA monomer. There are a lot of hydroxyl groups in complete alcoholysis PVA [16], and there are residual acetate groups in incomplete alcoholysis PVA, which has a great influence on the properties of PVA [17]. Zhao Caixia et al. [18]. Studied the effect of alcoholysis degree on the formation and properties of polyvinyl alcohol hydrogel. The results showed that the temperature sensitivity and dynamic rheological properties of PVA hydrogels with different alcoholysis degrees were also different. Huang Xiuling et al. [19] prepared PVA and thymol slow-release composite films, and they found that different alcoholysis degrees of PVA have a great influence on thymol release. 

The research on PVA as the matrix of composite materials not only studies the influence of PVA structure [20,21] but also studies the interface compatibility of PVA composites. The interface compatibility directly affects the functionality of the composites [22]; for example, Ji Jianchao et al. [23] studied the interface characteristics of polyvinyl alcohol (PVA) film and carbon powder for printing. By studying the influence of different alcoholysis degrees of PVA on the interface interaction of materials, it is found that the more appropriate PVA type makes the film and carbon powder have better compatibility and results in a better printing effect. Jawalkar et al. [24] believe that the mixed ratio of poly-L-lysine (PLL)/PVA has a great influence on the compatibility or incompatibility trend of composite materials, which leads to some limitations in the application of biotechnology. Therefore, they use molecular dynamics simulation to calculate the internal energy density of PLL/PVA with different mixing ratios and Flory–Huggins parameters, and they explain the relationship between the interface compatibility and molecular structure. Hailong Xu et al. [25] prepared the porous f-Ti_2_CT_x_/PVA foam by the freeze-drying process, and it possesses excellent EM absorption ability. This lightweight absorption-dominated electromagnetic interference (EMI) shielding materials minimize the twice pollution of the reflected electromagnetic (EM) wave and have a broad application prospect. Tatiana et al. [26] prepared composite films containing polyvinyl alcohol with different amounts of graphene oxide and modified the composite films with glycerin. The ductile properties of graphene oxide-filled polyvinyl alcohol composite films have increased significantly by using glycerin. However, all of the above studies are limited to one-sided macroscopic research and computer molecular scale research, and few researchers have provided a quantitative discussion on the compatibility parameters of PVA with different alcoholysis degrees. 

In this paper, the solubility parameters and interaction parameters (important parameters to measure the interfacial compatibility) of two kinds of commonly used PVA with different alcoholysis degrees were studied, and we calculated the data of part of the surface energy to transparentize the physical parameters such as the surface of the PVA. The solubility parameters, thermodynamic parameters, and some surface properties of PVA were calculated indirectly by the inverse gas chromatography (IGC) method and computer simulation. At the same time, the accuracy of solubility parameters obtained by the IGC method is calculated and verified by using the Forcite module of computer simulation software (Materials Studio 8.0, San Diego, CA, USA). Materials Studio is a material calculation software that can help researchers construct and analyze structural models of materials and predict the relevance properties of materials.

This paper provides a quantitative reference for future research on the relationship between the surface characteristics and structure of PVA composites by using the IGC method and computer simulation to calculate the relevant parameters of PVA with different alcoholysis degrees. It can also help other researchers choose the appropriate type of PVA when studying the PVA composites.

## 2. Materials and Methods

### 2.1. Raw Materials and Reagents

The experimental drugs and instruments used in this study are listed in Table 1.

#### Preparation of PVA Powder

The particles of PVA2488 and PVA2499 were crushed by a pulverizer, and the powder of more than 180 mesh was selected and put into a wide-mouth bottle for standby. The difference of PVA2488 and PVA2499 is that they have different alcoholysis degrees; the alcoholysis degree of PVA2488 is 88, while that of PVA2499 is 99.

### 2.2. IGC Determination of PVA Solubility Parameters and Surface Properties

PVA2488 was dissolved in acetone, and then, 6201 pickled red vectors were added to the PVA solution (volume of the discs was 1–1.2 times that of acetone) at a ratio of 1:10 (*w*/*w*)). The dry mixture was filled in a stainless-steel column having a length of 1.0 m and an inner diameter of 1/8 inch. The column was connected to an Agilent 6890 N gas chromatogram, and the column was aged for 24 h with N_2_ as the protection gas at 180 °C column temperature. After the aging, an IGC test could be carried out. A Hamilton syringe was used to inject 0.4 μL of probe solvent by hand, and three parallel injections were performed. The retention time was recorded. The solubility parameters and surface parameters of PVA were calculated according to the IGC theoretical formula. Test temperature: 110 °C, 120 °C, 130 °C, 140 °C, 150 °C.

### 2.3. Molecular Dynamics Simulation to Calculate the Solubility Parameters of PVA

PVA is a kind of copolymer obtained by the free radical polymerization of vinyl acetate, which is formed by the alcoholysis of the acetate group located along the PVAc chain. Since the reaction process is incomplete, it usually has two unit structures, as shown in Figure 1. According to the percentage of hydroxyl group in the product after alcoholysis, it is divided into partial alcoholysis (alcoholysis degree 80.0–98.5%), high alcoholysis (alcoholysis degree > 98.5%), and complete alcoholysis (alcoholysis degree 100.0%) [27,28].

#### 2.3.1. Determine the PVA Chain Length

A PVA chain was generated by the visualizer tool of MS software, and a PVA repeat unit was constructed by the homopolymer tool. As a result of the limited computing resources, it is very important to determine the appropriate PVA chain length. Based on the experience of many related researchers, this paper establishes a fully alcoholysis PVA long chain with repetition units of 10, 20, 30, 40, 50, and 60; then, it optimizes its structure by the Forcite module. The optimized PVA chain was used to establish its amorphous structure through amorphous cells (ACs). Finally, the solubility parameters of PVA with different chain lengths were calculated by the Forcite module, so as to determine the appropriate PVA chain length for further molecular dynamics research.

#### 2.3.2. Calculation of Solubility Parameters

Since PVA with an alcoholysis degree of 99 is close to complete alcoholysis, for the convenience of modeling, we use complete alcoholysis PVA (PVA_100_) instead of PVA with an alcoholysis degree of 99 to study. PVA with an alcoholysis degree of 88 (PVA_88_) was established by changing the occupancy ratio of two kinds of repeat units, and the single-chain models of PVA_88_ and PVA_100_ were iterated 5000 times by the steepest descent method and conjugate gradient method respectively to obtain the lowest energy. Then, the single-chain structure was annealed several times by the annealing program in the Forcite module, which was heated from 290 to 590 K at 50 K intervals and cooled back at 20 K intervals until the energy was no longer changed. Through this process, the energy balance of the system was accelerated. Then, the optimized PVA chain was used to establish its amorphous structure through amorphous cells (ACs). In order to avoid the ring-pricking phenomenon of a PVA long chain, the lower initial density of 0.6 g/cm^−3^ was set, and the smart algorithm was used to optimize its geometry for 10,000 iterations to obtain the equilibrium system. The AC amorphous equilibrium system was simulated by molecular dynamics (MD) [29,30,31]. First, the NVT ensemble was used to calculate the equilibrium of 100 ps, relax the structure, and release the tension of the system to achieve the equilibrium of geometric conformation; then, 400 ps was balanced under the NPT ensemble, and then, 100ps was run through NVT, and the trajectory file was analyzed to calculate the solubility parameters of two alcoholysis degrees at 298.15 K. (The force field selection for all molecular dynamics simulation processes: COMPASSII Version 1.2, temperature control method: nose, pressure control method: Andersen, step length 1fs, accuracy: 0.001 kcal/mol, cutoff distance: 12.5 A.)

## 3. Results and Discussion

### 3.1. IGC Method to Calculate the Solubility Parameters of PVA

#### 3.1.1. Characteristic Retention Volume of Probe Solvent

Using the IGC method [32], the probe solvent retention volume $V_{g}^{0}$ can be obtained by Formulas (1) and (2). The specific values are shown in Table 2.
(1)$${\;V}_{g}^{0}=273.15\mathrm{JF}\frac{\Delta t}{\mathrm{mT}}$$

Among them, J is the non-ideal gas compression factor, F is the carrier velocity, Δt is the difference between the retention time t_r_ of the probe solvent and n-pentane t_m_, m is the mass of the fixed-phase PVA, and T is the column temperature.
(2)$$J=\frac{3}{2}\frac{{\left(P_{i}/P_{o}\right)}^{2}-1}{{\left(P_{i}/P_{o}\right)}^{3}-1}$$

P_i_ and P_o_ are the inlet and outlet pressure of the stainless-steel column.

The retention volumes of probe solvents on PVA2488 and PVA2499 at different temperatures are listed in Table 2. On the whole, the increase of temperature destroys and weakens the interaction between the probe solvent and PVA, so the interaction time between the probe solvent and PVA decreases, and the retention volume of the probe solvent and PVA decreases; in homologues, the larger the molecular weight of the probe solvent is, the stronger the interaction between the probe solvent and PVA, and the larger the retention volume. For different kinds of probe solvents, the retention time of some probe solvents (ethanol, 1-propanol, tetrahydrofuran, ethyl acetate, acetonitrile) on PVA was longer than other probe solvents at the same temperature. The larger retention volume of ethanol and 1-propanol may be due to the formation of intermolecular hydrogen bonds between small alcohols containing more hydroxyl groups and PVA; tetrahydrofuran (THF), ethyl acetate, and acetonitrile are the most polar solvents in common mobile phases, which may interact with PVA with strong polarity, resulting in the increase of retention volume.

Comparing the retention volumes of PVA2488 and PVA2499, it can be seen that the retention volumes of PVA2488 and alkane probe solvents are larger than those of PVA2499, but the retention volumes of PVA2499 and non-alkane probe solvents are larger than those of PVA2488, which may be due to the fact that PVA2499 has more hydroxyl groups and weaker interaction with alkane non-polar probe solvents than other solvents.

#### 3.1.2. Thermodynamic Parameters of Probe Molecule

From the $V_{g}^{0}$ of the probe solvent, the molar absorption enthalpy ${\Delta H}_{l}^{s}$, the infinite dilution molar mixing enthalpy ${\Delta H}_{l}^{\infty }$, and the molar evaporation enthalpy ${\Delta H}_{v}$ can be calculated [33,34]. The calculation Formulas (3)–(5) are as follows:(3)$${\Delta H}_{l}^{s}=-R\frac{\partial \left(\ln V_{g}^{0}\right)}{\partial \left(1/T\right)}$$
(4)$${\Delta H}_{l}^{\infty }=R\frac{\partial \left(\ln\Omega_{1}^{\infty }\right)}{\partial \left(1/T\right)}$$
(5)$${\Delta H}_{v}={\Delta H}_{l}^{\infty }-{\Delta H}_{l}^{s}$$
where $\Omega_{1}^{\infty }$ is the infinite dilution activity coefficient [35], which is obtained by Equation (6):(6)$$\ln\Omega_{1}^{\infty }=\ln\frac{273.15R}{P_{1}^{0}V_{g}^{0}M_{1}}-\frac{P_{1}^{0}}{\mathrm{RT}}\left(B_{11}-V_{1}\right)$$
where R is the universal gas constant, $M_{1}$ is the molar mass of the probe solvent, and $P_{1}^{0}$, $B_{11}$, and $V_{1}$ are the saturated vapor pressure at the temperature of the probe solvent, the second virial coefficient, and the molar volume, respectively. The specific values are shown in Table 3 and Table 4.

The values of ${\Delta H}_{l}^{s}$, ${\Delta H}_{l}^{\infty }$, and ${\Delta H}_{v}$ of PVA2488 and PVA2499 are presented in Table 3. The thermodynamic parameters were compared between PVA2488 and PVA2499 in the same kind of probes, and there was no significant rule. However, in the homologues, the molar evaporation enthalpy ${\Delta H}_{v}$ values increased with the increasing carbon number, which indicated that the higher energy was required for the evaporation of the probe solvent from PVA, which was associated with stronger PVA adsorption.

#### 3.1.3. Flory–Huggins Interaction Parameters${\;\chi }_{12}^{\infty }$

The Flory–Huggins interaction parameter ${\;\chi }_{12}^{\infty }$ reflects the change of interaction energy when PVA is mixed with the probe solvent [36], and the smaller the ${\;\chi }_{12}^{\infty }$ value, the stronger the dissolving ability of the probe solvent to PVA, which can be calculated by Equation (7), and the results are shown in Table 4.
(7)$$\chi_{12}^{\infty }=\ln\frac{273.15R}{P_{1}^{0}V_{g}^{0}M_{1}}-\frac{P_{1}^{0}}{\mathrm{RT}}\left(B_{11}-V_{1}\right)-1$$

Table 4 shows that the Flory–Huggins interaction parameter ${\;\chi }_{12}^{\infty }$ between the probe solvent and PVA at 383.15–423.15 K is less than 0.5. The results show that these probes are good solvents for PVA2488. In PVA2499, the ${\;\chi }_{12}^{\infty }$ values of butanone, p-xylene, o-xylene, ethylbenzene, ether, tetrahydrofuran, ethyl acetate, and acetonitrile are all less than 0.5, indicating that these probes are good solvents for PVA2499, and they provide comparable solvents. The lower ${\;\chi }_{12}^{\infty }$ value of PVA2499 indicates that PVA2499 is more soluble in these solvents.

#### 3.1.4. Solubility Parameters

The solubility parameter $\delta_{1}$ of the probe solvent is expressed by Formula [37] (8):(8)$$\delta_{1}={\left(\frac{{\Delta H}_{v}-\mathrm{RT}}{V_{1}}\right)}^{\frac{1}{2}}$$

The solubility parameters $\delta_{2}$ of PVA can be obtained by using $\delta_{1}$ of the probe solvent and Formula (9).
(9)$$\frac{\delta_{1}^{2}}{\mathrm{RT}}-\frac{\chi_{12}^{\infty }}{V_{1}}=\left(\frac{2\delta_{2}}{\mathrm{RT}}\right)\delta_{1}-\left(\frac{\delta_{2}^{2}}{\mathrm{RT}}+\frac{\chi_{s}^{\infty }}{V_{1}}\right)$$
where $\chi_{S}^{\infty }$ is the entropy term of the Flory–Huggins interaction parameter.

The calculated results of solubility parameter $\delta_{2}$ of PVA at 383.15–423.15 K are shown in Table 5. These $\delta_{2}$ values can be obtained by plotting $\delta_{1}^{2}/\mathrm{RT}-\frac{\chi_{12}^{\infty }}{V_{1}}$ and $\delta_{1}$ (as shown in Figure 2), and the solubility parameter of PVA at 298.15 K at room temperature can be obtained by extrapolation (as shown in Figure 3).

Since the IGC method needs to vaporize the solvent, the solubility parameter at 298.15 K cannot be measured, so extrapolation is required. By measuring the solubility parameters at 383.15 K, 393.15 K, 403.15 K, 413.15 K, and 423.15 K, using the origin software for linear fitting, the linear relationship between the solubility parameters of PVA2488 and PVA2499 and the temperature was obtained, and the solubility parameter at 298.15 K was obtained.

It can be seen from Table 5 that the solubility parameters of PVA2488 and PVA2499 at room temperature are 23.15 and 25.62 (J/cm^−3^)^0.5^, respectively. PVA2499 has a higher solubility parameter, which is probably due to the fact that PVA2499 has more -OH groups, and compared with -COOCH_3_ groups, -OH has a greater contribution to the solubility parameter delta, and PVA2499 has more hydrophilic hydroxyl groups, forcing a large number of hydrogen bonds between molecules, which makes the cohesive force of the polymer chain stronger, thus making the solubility parameter of PVA2499 larger.

### 3.2. Calculation of Solubility Parameters by Molecular Dynamics Simulation

The trend diagram of the PVA repeat unit numbers and solubility parameters is shown in Figure 4a. It can be seen from Figure 4 and Table 6 that the solubility parameter difference of PVA with a repeat unit number of 30 or more chain length is small, so the PVA chain with a repeat unit number of 30 or more can meet the requirements of calculation accuracy. As a result of the long chain length, the calculation speed is slow. In order to improve the calculation efficiency and make the repeat unit ratio of the PVA_88_ model reasonable, the PVA chain with 50 repeat units is selected for molecular dynamics simulation. The random stereoscopic model of PVA_100_ with a repeat unit n = 50 and the random stereoscopic model of PVA_88_ with a repeat unit ratio m:n = 6:44 (see Figure 4b) were established by the copolymer tool. The amorphous structure system (see Figure 4c) was established by the model, and then, the molecular dynamics simulation calculation was carried out. The molecular dynamics calculation process is the same as that in Section 2.3.2. From the final NVT output stage, we can evaluate whether the system reaches equilibrium. The fluctuation of temperature and energy distribution in a small range can be considered as an equilibrium system (see Figure 4d), which can be used to calculate the solubility parameters. From the figure, we can see that the fluctuation of energy and temperature is very small, which indicates that the system has reached equilibrium.

It can be seen from Table 7 that the value of PVA100 is larger than that of PVA88, and the van der Waals effect has a greater contribution to the total solubility parameter of PVA100, which may be due to the relatively large molecular weight of PVA88, which makes the molecular volume larger and the value of PVA88 smaller.

### 3.3. Comparison of Solubility Parameters between IGC Method and Molecular Dynamics Simulation of PVA

The solubility parameters of PVA with different alcoholysis degrees were studied by the IGC method and molecular dynamics method. The solubility parameters of PVA2488 and PVA2499 measured by the IGC method were 23.74 and 25.62 (J/cm^−3^)^0.5^, respectively. The solubility parameters of PVA_88_ and PVA_100_ calculated by molecular dynamics simulation were 22.68 and 24.36 (J/cm^−3^)^0.5^, respectively. 

The solubility parameters of PVA with different alcoholysis degrees calculated by the two methods are significantly similar; comparing the data measured by the IGC method and the PVA molecular dynamics method can prove the reliability of the IGC method, and it can prove that the PVA molecular dynamics method can also be used to calculate and verify the experimental results. Therefore, the PVA molecular dynamics method can also be used in other studies to verify or directly calculate the corresponding parameters, which is very helpful for some data that are difficult to measure.

### 3.4. Surface Properties

The surface energy ${\;\gamma }_{s}$ refers to the extra energy of the solid surface relative to the interior [38,39]. In this part, the surface energy of PVA with different alcoholysis degrees was calculated by IGC technology at the temperature of 383.15–423.15 K. The specific values are listed in Table 8.

#### 3.4.1. Dispersion Component $\gamma_{s}^{d}$


The surface energy dispersive component $\gamma_{s}^{d\;}$ represents the interaction force between the surface of solid materials and non-polar molecules [40], which can be calculated according to the Dorris–Gray method [41]. See Formulas (10) and (11):(10)$$\gamma_{s}^{d}=\frac{1}{\gamma_{{\mathrm{CH}}_{2}}}\cdot {\left(\frac{{\Delta G}_{{\mathrm{CH}}_{2}}}{2{\mathrm{Na}}_{{\mathrm{CH}}_{2}}}\right)}^{2}$$
(11)$$\gamma_{{\mathrm{CH}}_{2}}=36.8-0.058t$$
where n is the Avogadro constant; ${\Delta G}_{{\mathrm{CH}}_{2}}$ is the increment of adsorption free energy, which is the slope of the linear relationship between $\mathrm{RT}\ln V_{g}^{0}$ and the number of carbon atoms, as shown in Figure 5;${\;\gamma }_{{\mathrm{CH}}_{2}}\;$is the solid surface energy composed of a methylene group, and $a_{{\mathrm{CH}}_{2}}$ is the cross-sectional area of the methylene group, which is 0.06 nm^2^.

Figure 5 plots the linear graph of $\mathrm{RT}\ln V_{g}^{0}$ and the number of carbon atoms of n-alkanes corresponding to PVA. Its slope is ${\Delta G}_{{\mathrm{CH}}_{2}}$. Substituting it into Equation (10) can obtain the dispersion component $\gamma_{s}^{d\;}$of PVA. The lower the dispersion component $\gamma_{s}^{d\;}$ of PVA at 383.15–423.15 K, the more specific the value that can be obtained from Table 8, and in conjunction with Figure 6, it can be seen directly that the dispersion component $\gamma_{s}^{d\;}$does not change significantly with the increase in temperature, indicating that the dispersion effect between the nonpolar probe and the PVA surface under the temperature gradient of 383.15–423.15 K is less affected by temperature, the $\gamma_{s}^{d\;}$value is relatively close and less sensitive to temperature. This is most likely due to the too high temperature, which makes a certain degree of deformation on the PVA surface, so that the surface dispersion force is not only affected by the nonpolar functional groups. It may also have some relationship with the surface roughness of PVA, but due to the complexity of the situation, a more regular $\gamma_{s}^{d\;}$cannot be obtained, so the dispersion component $\gamma_{s}^{d\;}\;$of PVA at room temperature cannot be obtained by extrapolation. Comparing the $\gamma_{s}^{d\;}$of PVA2488 and PVA2499, PVA2499 has a higher $\gamma_{s}^{d\;}$at the same temperature.

#### 3.4.2. Specific Component $\gamma_{s}^{\mathrm{sp}}$ and Total Surface Energy $\gamma_{s}$

The surface-specific component $\gamma_{s}^{\mathrm{sp}}$ reflects the surface energy of the dispersive part between the filler and the probe, which includes polarity, acid–base, and hydrogen bonding. It can be calculated by the Good–van Oss method [42] combined with the specific adsorption free energy ${\Delta G}^{\mathrm{sp}}$. The formula is as follows:(12)$${\Delta G}^{\mathrm{sp}}=2\mathrm{Na}({\left(\gamma_{l}^{+}\gamma_{s}^{-}\right)}^{1/2}+\left(\gamma_{l}^{-}\gamma_{s}^{+}{)}^{1/2}\right)$$
(13)$$\gamma_{s}^{\mathrm{sp}}=2{\left(\gamma_{l}^{+}\gamma_{s}^{-}\right)}^{1/2}$$
(14)$$\gamma_{s}^{d}=\frac{1}{\gamma_{{\mathrm{CH}}_{2}}}\cdot {\left(\frac{{\Delta G}_{{\mathrm{CH}}_{2}}}{2{\mathrm{Na}}_{{\mathrm{CH}}_{2}}}\right)}^{2}.$$

The $\gamma_{l}^{+}$ and $\gamma_{l}^{-}$ [35] of dichloromethane are 124.58 and 0 mJ/m^2^, respectively, and the $\gamma_{l}^{+}$ and $\gamma_{l}^{-}$ of toluene are 16.23 and 0 mJ/m^2^, respectively. From this, the specific component and total surface energy of the filler can be calculated. The specific component $\gamma_{s}^{\mathrm{sp}}$ of the PVA in addition to the dispersion part is calculated using dichloromethane and toluene. The calculation results are shown in Table 8. In addition, the total surface energy of the PVA can be obtained by Formula (14).

It can be seen from Figure 6 that the specific component of PVA and the total surface energy and dispersion component do not change much with temperature, but in general, the specific component and total surface energy of PVA2499 are larger than those of PVA2488. It may be due to the fact that PVA2499 has more exposed active hydroxyl groups than PVA2488 under the same degree of polymerization, which makes it have higher surface energy. It is likely that PVA2488, which has a lower degree of alcoholysis, contains more residual acetate and is a hydrophobic group, which reduces the tension of the probe molecule on its surface, so it has a lower surface energy than PVA2499.

#### 3.4.3. Surface Acidity and Alkalinity

In addition to the dispersion effect on the polar probe molecule, the surface of PVA also has an acid–base effect. According to the Sawyer method [43,44], Formula (15) can be used to obtain the surface-specific adsorption free energy $\Delta G^{\mathrm{sp}}$ of PVA at different temperatures. Figure 7 shows the calculation diagram of the acidic probe chloroform, the basic probe tetrahydrofuran, and the amphoteric probe ethyl acetate on the surface $\Delta G^{\mathrm{sp}}\;$of the PVA calculation diagram; the straight line in the figure is the relationship between the $\mathrm{RT}\ln V_{g}^{0}\;$of n-alkane and the boiling point t_b_ of the probe solvent. This straight line is a reference, and the distance between the unipolar probe and the amphipathic probe that deviates from this straight line is the $\Delta G^{\mathrm{sp}}$ of the probe on the PVA surface.
(15)$${\Delta G}^{\mathrm{sp}}={\mathrm{RTlnV}}_{g}^{0}-{\mathrm{RTlnV}}_{g}^{\mathrm{ref}}$$

Table 9 lists the characteristic adsorption free energies $\Delta G^{\mathrm{sp}}$ of different probes on the surface of PVA2488 and PVA2499. Overall, tetrahydrofuran, trichloromethane, as well as ethyl acetate are all located above the straight line, illustrating the existence of both acidic and basic sites on the surface of PVA, which are amphoteric materials. Since the $\Delta G^{\mathrm{sp}}$ value of basic probe tetrahydrofuran on the surface of PVA is larger relative to the acidic probe trichloromethane, this indicates that the PVA surface has a stronger acidic effect; that is, PVA is a zwitterionic acid material; from the table, the characteristic adsorption free energy of the two PVAs can be compared. At the same temperature, the $\Delta G^{\mathrm{sp}}\;$of trichloromethane on the surface of PVA2488 is larger and that of tetrahydrofuran on PVA2499 is larger, which also indicates that PVA2499 is relatively more acidic with respect to the PVA2488 surface.

## 4. Conclusions

In order to choose the suitable PVA for the subsequent composites, in this paper, the solubility parameters of PVA with two alcoholysis degrees were studied by the IGC method, aiming to provide a quantitative reference for the subsequent interfacial compatibility study, and a PVA model corresponding to two alcoholysis degrees was built by molecular dynamics computer simulation and solubility parameter calculation, which verified the experimental calculation by the IGC method, and the results showed that the two inter-rater calculations had consistency. At the same time, the surface energy and surface energy components (dispersive surface energy component${\;\gamma }_{s}^{d}$, specific surface energy component${\;\gamma }_{s}^{\mathrm{sp}}$) are calculated with the help of the IGC method. The results show that the dispersion of the nonpolar probe and the PVA surface is affected by temperature at a temperature above 110 °C. The dispersion component${\;\gamma }_{s}^{d}$ of PVA at room temperature cannot be obtained by extrapolation, but the dispersion component contributes a lot to the surface energy as a whole, and compared to PVA2499, the surface energy of PVA2488 is larger. In addition, the surface acid–base properties of PVA with two alcoholysis degrees were investigated at a certain temperature, and the results showed that PVA was an amphoteric acid material, and the surface with a larger alcoholysis degree (PVA2499) had strong acidity.

This paper, through the study of PVA’s surface properties with solubility parameters, as well as interaction parameters, and so on, can provide a valuable reference for the relationship between its structure and properties of composites, so as to make the corresponding composites have better interfacial bonding properties, for example, improving the performance of PVA hydrogel and PVA composite film.

In addition, the MD method used in this article can not only calculate the parameters of the surface properties of PVA but also simulate and calculate other data; for example, many studies have shown that the glass transition temperature (TG) of the material can be derived from the temperature–volume diagram obtained by MD. In the following research, we will continue to study the molecular dynamics in depth and explore the simulation calculation of other parameters such as TG by this method.

## Figures and Tables

**Figure 1 polymers-13-03778-f001:**
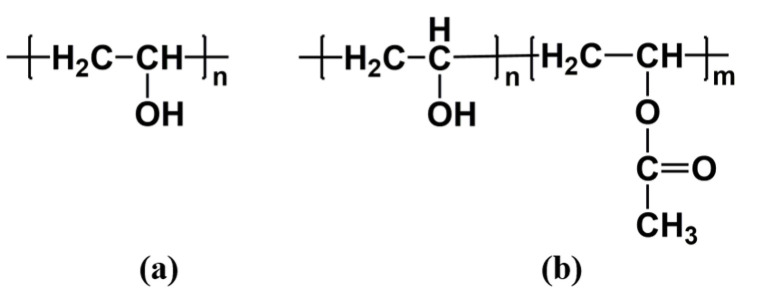
Chemical structure: (**a**) fully hydrolyzed PVA, (**b**) partially hydrolyzed PVA.

**Figure 2 polymers-13-03778-f002:**
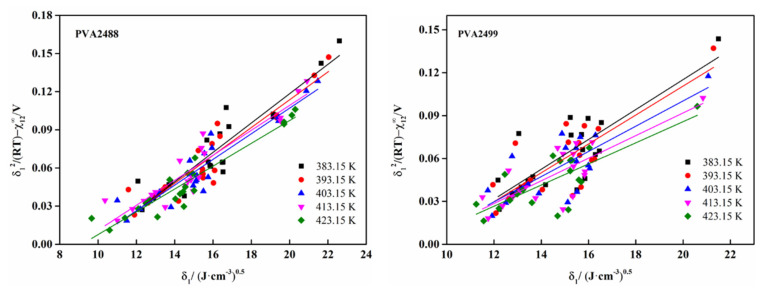
$\frac{\delta_{1}^{2}}{\mathrm{RT}}-\frac{\chi_{12}^{\infty }}{V_{1}}$ and ${\;\delta }_{1}$ curves of PVA at different temperatures.

**Figure 3 polymers-13-03778-f003:**
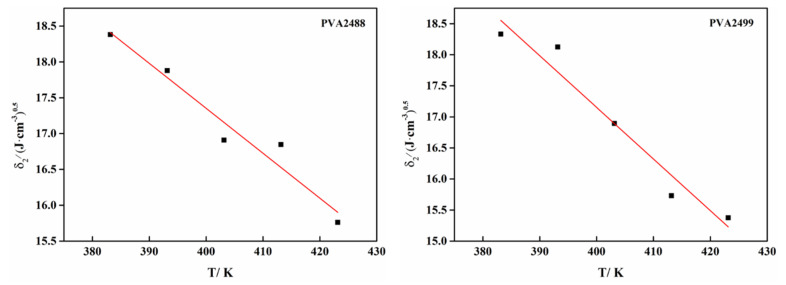
Calculation of PVA solubility parameter (δ_2_) at 298.15 K by extrapolation.

**Figure 4 polymers-13-03778-f004:**
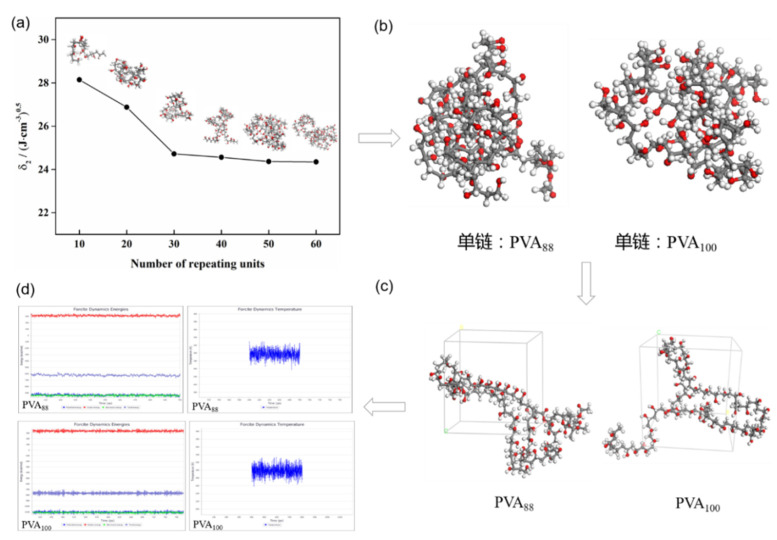
(**a**) The relationship between the number of repeat units and solubility parameters of PVA (298.15 K); (**b**) single-chain model of PVA with two alcoholysis degrees; (**c**) amorphous structure of PVA with two alcoholysis degrees; (**d**) energy and temperature distribution of PVA with two alcoholysis degrees.

**Figure 5 polymers-13-03778-f005:**
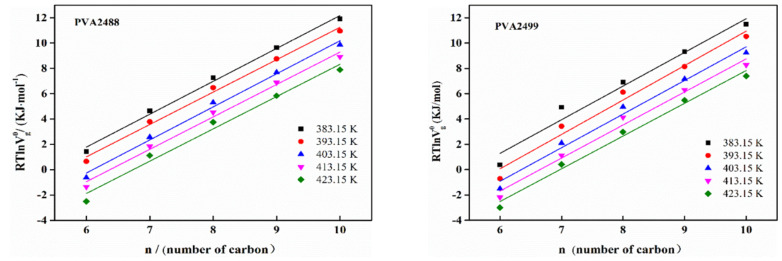
Variation of $\mathrm{RT}\ln V_{g}^{0}$ with the number of carbon atoms of n-alkanes at different temperatures.

**Figure 6 polymers-13-03778-f006:**
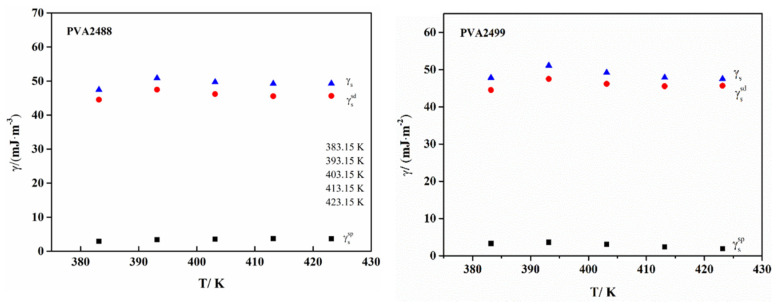
Surface energy of PVA2488 and PVA2499 at different temperatures.

**Figure 7 polymers-13-03778-f007:**
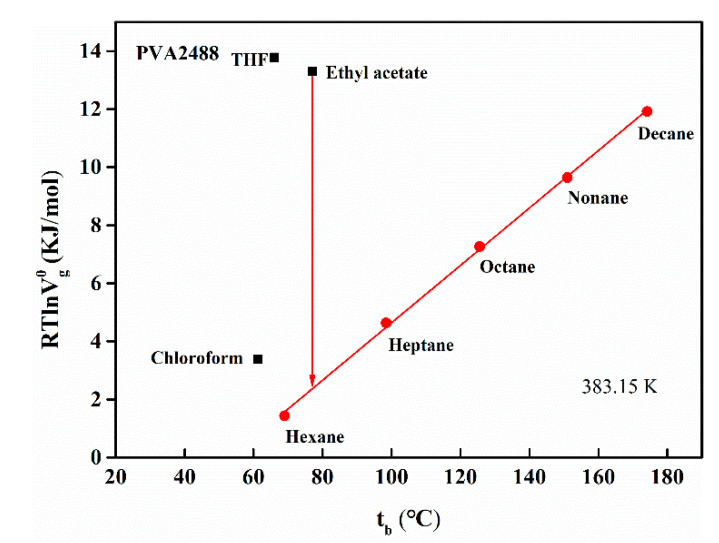
The relationship between $\mathrm{RT}\ln V_{g}^{0}$ and the boiling point${\;t}_{b}$ of the probe solvent (at 383.15 K, take PVA2488 as an example).

**Table 1 polymers-13-03778-t001:** Experimental drugs and instruments.

Name	Purity or Model	Manufacturer
PVA2488	analytical purity	Shanghai Yousuo Chemical Technology Co., Ltd. (Shanghai, China)
PVA2499	daily chemical grade	Guangzhou Suixin Chemical Co., Ltd. (Guangzhou, China)
Huangcheng pulverizer	HC-2000Y	Yongkang Tianqi Shengshi industry and Trade Co., Ltd. (Huangcheng, China)
6201 Pickling red carrier	60–80 mesh	Tianjin Guangfu Fine Chemical Research Institute Co., Ltd. (Tianjin, China)
Probe solvents	all chromatographic purity	Aladdin reagent Co., Ltd. (Shanghai, China)
Gas chromatograph	6890 N	Agilent Technologies Co., Ltd. (Beijing, China)

**Table 2 polymers-13-03778-t002:** Specific retention volumes at different temperatures${\;V}_{g}^{0}$ (ml/g).

Probe Solvent	PVA2488					PVA2499				
	383.15 K	393.15 K	403.15 K	413.15 K	423.15 K	383.15 K	393.15 K	403.15 K	413.15 K	423.15 K
n-hexane	1.57	1.22	0.83	0.67	0.49	1.13	0.80	0.64	0.53	0.43
n-heptane	4.28	3.19	2.15	1.71	1.38	4.70	2.86	1.87	1.38	1.12
n-octane	9.77	7.26	4.86	3.74	2.90	8.74	6.52	4.38	3.34	2.33
n-nonane	20.63	14.57	9.89	7.42	5.26	18.62	12.06	8.47	6.27	4.77
n-decane	42.11	28.57	19.01	13.44	9.43	36.81	25.05	15.75	11.11	8.18
cyclohexane	1.57	1.22	0.94	0.73	0.59	1.27	0.94	0.77	0.57	0.43
dichloromethane	1.75	1.22	0.99	0.83	0.64	1.69	1.25	0.89	0.61	0.43
trichloromethane	2.90	2.26	1.82	1.50	1.23	1.88	1.12	0.85	0.65	0.27
trichloroethylene	4.22	3.13	2.38	1.87	1.38	3.67	2.68	2.04	1.42	0.74
ethanol	58.40	40.25	17.85	11.94	9.19	-	-	-	-	-
1-propanol	-	-	-	-	-	112.50	67.25	39.33	14.89	8.96
acetone	44.70	34.40	21.22	8.46	7.17	60.27	37.16	21.54	14.73	11.17
ketone	66.42	44.24	29.12	19.62	11.84	112.36	75.74	38.40	28.20	8.96
benzene	8.14	6.25	4.64	3.63	2.95	-	-	-	-	-
toluene	13.03	9.50	6.46	4.93	3.78	15.09	9.96	6.13	4.27	3.10
p-xylene	29.38	20.56	13.54	9.50	7.22	30.75	20.86	13.83	8.71	6.28
o-xylene	33.96	24.15	17.18	10.85	8.30	35.35	22.24	16.09	11.07	6.75
ethylbenzene	28.59	18.53	13.04	8.98	6.68	33.47	18.31	13.37	8.99	5.55
ether	27.39	18.07	9.45	6.28	3.64	36.77	26.97	17.79	10.42	5.97
THF	75.53	38.86	26.19	15.10	9.53	108.84	64.80	30.10	12.62	10.82
methyl ethyl	65.15	41.12	25.42	15.10	11.25	87.68	68.15	39.72	15.30	10.12
acetonitrile	67.75	49.11	30.89	21.48	15.23	80.58	62.43	21.88	9.69	7.60

**Table 3 polymers-13-03778-t003:** Thermodynamic parameters of the probe solvent (KJ/mol).

Probe Solvent	PVA2488			PVA2499		
	$\Delta H_{l}^{s}$	$\Delta H_{l}^{\infty }$	$\Delta H_{v}$	$\Delta H_{l}^{s}$	$\Delta H_{l}^{\infty }$	$\Delta H_{v}$
n-hexane	−39.48	−13.64	25.84	−31.95	−5.79	26.16
n-heptane	−46.02	−15.65	30.37	−48.55	−17.88	30.67
n-octane	−42.29	−7.56	34.73	−44.70	−9.66	35.03
n-nonane	−46.51	−7.51	39.00	−45.64	−6.32	39.32
n-decane	−50.43	−7.18	43.25	−51.57	−7.98	43.59
cyclohexane	−37.40	−9.69	27.71	−36.08	−14.58	21.50
dichloromethane	−29.01	−8.01	21.00	−46.74	−19.93	26.81
trichloromethane	−31.88	−5.81	26.07	−59.19	−28.86	30.34
trichloroethylene	−35.69	−5.85	29.83	−51.50	−22.61	28.89
ethanol	−65.38	−28.93	36.45	-	-	-
1-propanol	-	-	-	−88.35	−49.82	38.53
acetone	−65.41	−39.09	26.32	−58.08	−20.48	37.60
ketone	−55.77	−26.52	29.25	−81.09	−51.50	29.59
benzene	−35.35	−6.93	28.42	-	-	-
toluene	−42.26	−9.78	32.48	−54.13	−15.41	38.72
p-xylene	−48.72	−8.38	40.35	−54.58	−24.21	30.37
o-xylene	−48.49	−14.57	33.92	−53.99	−31.81	22.19
ethylbenzene	−48.57	−12.05	36.51	−58.08	−29.83	28.25
ether	−68.36	−46.65	21.71	−61.58	−31.93	29.65
THF	−65.08	−46.25	18.83	−84.45	−65.21	19.24
methyl ethyl	−63.32	−34.01	29.31	−78.05	−56.56	21.49
acetonitrile	−51.91	−21.19	30.71	−88.80	−58.40	30.40

**Table 4 polymers-13-03778-t004:** Flory–Huggins interaction parameters of probe solvent and PVA at different temperatures.

Probe Solvent	PVA2488					PVA2499				
	383.15 K	393.15 K	403.15 K	413.15 K	423.15 K	383.15 K	393.15 K	403.15 K	413.15 K	423.15 K
n-hexane	3.09	3.12	3.30	3.33	3.47	3.42	3.54	3.57	3.57	3.61
n-heptane	2.70	2.74	2.89	2.91	2.92	2.60	2.85	3.04	3.12	3.12
n-octane	2.50	2.51	2.64	2.66	2.67	2.61	2.62	2.75	2.77	2.89
n-nonane	2.38	2.41	2.50	2.50	2.59	2.49	2.60	2.65	2.67	2.68
n-decane	2.31	2.34	2.41	2.45	2.51	2.44	2.47	2.60	2.64	2.66
cyclohexane	3.41	3.44	3.48	3.54	3.56	2.62	2.74	2.78	2.92	3.07
dichloromethane	2.13	2.29	2.30	2.32	2.42	2.78	2.86	2.99	3.18	3.36
trichloromethane	1.92	1.95	1.96	1.97	2.00	3.00	3.27	3.31	3.36	4.03
trichloroethylene	2.10	2.16	2.20	2.23	2.33	2.06	2.14	2.19	2.34	2.81
ethanol	0.03	0.10	0.63	0.77	0.79	-	-	-	-	-
1-propanol	-	-	-	-	-	1.21	1.42	1.66	2.35	2.60
acetone	−0.30	−0.26	0.02	0.75	0.74	1.69	1.87	2.13	2.24	2.26
ketone	−0.28	−0.11	0.08	0.26	0.57	−0.81	−0.65	−0.20	−0.10	0.85
benzene	1.80	1.83	1.90	1.95	1.96	-	-	-	-	-
toluene	1.97	2.02	2.15	2.19	2.24	1.38	1.48	1.67	1.75	1.82
p-xylene	2.00	2.05	2.18	2.26	2.28	0.11	0.23	0.41	0.66	0.79
o-xylene	1.68	1.72	1.78	1.98	2.00	−0.44	−0.16	−0.01	0.21	0.56
ethylbenzene	1.78	1.92	1.99	2.10	2.15	0.08	0.45	0.55	0.75	1.04
ether	−0.48	−0.25	0.22	0.48	0.88	0.20	0.27	0.46	0.78	1.14
THF	−1.84	−1.34	−1.11	−0.68	−0.34	−2.20	−1.85	−1.23	−0.50	−0.47
methyl ethyl	−0.53	−0.31	−0.06	0.26	0.36	−1.79	−1.72	−1.34	−0.54	−0.27
acetonitrile	0.29	0.36	0.59	0.73	0.87	0.09	0.10	0.91	1.51	1.55

**Table 5 polymers-13-03778-t005:** Solubility parameters $\delta_{2}$ of PVA at different temperatures ((J/cm^−3^)^0.5^).

Model	383.15 K	393.15 K	403.15 K	413.15 K	423.15 K	298.15 K
**PVA2488**	1838	17.88	16.91	16.85	15.76	23.74
**PVA2499**	18.33	18.12	16.89	15.73	15.37	25.62

**Table 6 polymers-13-03778-t006:** Solubility parameters of PVA with different alcoholysis degrees simulated by molecular dynamics (at 298.15 K).

Repeat Units	10	20	30	40	50	60
**PVA_100_**	28.15	26.88	24.72	24.56	24.36	24.35
**PVA_88_**	-	-	-	-	22.68	-

**Table 7 polymers-13-03778-t007:** Solubility parameter components and total solubility parameters of PVA with two alcoholysis degrees at 298.15 K.

Model	$\delta_{elec}/(J\cdot {\mathrm{cm}}^{-3}{)}^{0.5}$	$\delta_{vdw}/(J\cdot {\mathrm{cm}}^{-3}{)}^{0.5}$	$\delta_{total}/(J\cdot {\mathrm{cm}}^{-3}{)}^{0.5}$
**PVA_88_**	16.03	15.75	22.68
**PVA_100_**	14.53	19.27	24.36

**Table 8 polymers-13-03778-t008:** Dispersive component $\gamma_{s}^{d}$, specific component $\gamma_{s}^{d}$, and total surface energy $\gamma_{s}$ of PVA2488 and PVA2499 at different temperatures.

T/K	$\gamma_{s}^{d}$	$\gamma_{s}^{-}$	$\gamma_{s}^{+}$	$\gamma_{s}^{\mathrm{sp}}$	$\gamma_{s}$
PVA2488					
383.15	42.48	0.88	1.25	2.10	44.58
393.15	42.02	1.35	2.22	3.46	45.48
403.15	44.64	1.04	1.35	2.37	47.01
413.15	43.75	1.05	1.34	2.37	46.13
423.15	44.37	1.05	1.50	2.50	46.87
PVA2499					
383.15	44.53	1.19	2.28	3.29	47.82
393.15	47.47	1.43	2.27	3.60	51.08
403.15	46.15	1.28	1.82	3.05	49.20
413.15	45.53	0.90	1.53	2.35	47.88
423.15	45.63	0.63	1.37	1.86	47.49

**Table 9 polymers-13-03778-t009:** Surface characteristics of PVA2488 and PVA2499 at different temperatures adsorption free energy $\Delta G^{\mathrm{sp}}$ (KJ/mol).

**PVA2488**					
Probe solvent	383.15 K	393.15 K	403.15 K	413.15 K	423.15 K
Chloroform	2.58	2.61	3.26	3.32	3.57
THF	12.50	11.45	11.72	10.78	10.33
Ethyl acetate	10.94	10.57	10.52	9.71	9.84
**PVA2499**					
Probe solvent	383.15 K	393.15 K	403.15 K	413.15 K	423.15 K
Chloroform	1.74	1.33	1.39	1.21	−1.09
THF	14.19	14.12	12.86	10.92	11.41
Ethyl acetate	12.39	13.14	12.68	10.49	10.09

## Data Availability

All the data analyzed in this article are obtained during the experiment and are presented in the form of a table. We agree to share these data.

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
