# Peer review of "Quantitative Study on Solubility Parameters and Related Thermodynamic Parameters of PVA with Different Alcoholysis Degrees"

_polymers, 2021, doi:10.3390/polym13213778_

Round 1

Reviewer 1 Report

Title: Quantitative study on solubility parameters and surface properties of PVA with different alcoholysis degrees

Authors: Siqi Chen, Hao Yang, Kui Huang, Xiaolong Ge, Hanpeng Yao, Junxiang Tang, Junxue Ren, Shixue Ren, Yanli Ma

Journal: Polymers
Manuscript ID: polymers-1402432

Review:

In this paper, using inverse gas chromatography experiment, some parameters of polyvinyl alcohol (PVA) with two different alcohol degrees are determined, including solubility parameters and the surface properties (like surface energy and acid-base property). In parallel, solubility parameters are obtained by molecular dynamics simulation (MD) using the moduli in Materials Studio software and the results are compared. Also, a trade-off on the chain length is conducted in MD simulation to converge the result of atomistic simulation to that of the experiment.

The overall idea of atomistic simulation-IGC experiment and the effort for the convergence of the results seems interesting and worth publishing. However, the manuscript needs to go under a severe editorial revision as it contains numerous errors and also the following remarks needs to be addressed:

Major Remarks:

  • The temperature at which the experimental solubility parameter is compared with atomistic simulation is obtained through extrapolation rather than direct measurements while this is the main value for verification of the method. Authors should provide explanation if it relates to any limitation in the experiment.
  • For relaxation of amorphous polymer unit cells (besides the single polymer chain) in MD simulation, it is recommended to perform annealing above the glass transition temperature where the higher mobility of polymer chains makes it easier to achieve equilibration within reasonable computational time. For a complete relaxation procedure of polymer unit cells, the Authors are referred to “Izadi, R., Ghavanloo, E., & Nayebi, A. (2019). Elastic properties of polymer composites reinforced with C60 fullerene and carbon onion: Molecular dynamics simulation. Physica B: Condensed Matter574, 311636.” and “Malagù, M., Goudarzi, M., Lyulin, A., Benvenuti, E., & Simone, A. (2017). Diameter-dependent elastic properties of carbon nanotube-polymer composites: Emergence of size effects from atomistic-scale simulations. Composites Part B: Engineering131, 260-281.”
  • The Materials Studio software used for the MD simulation should be clearly mentioned and referenced in the manuscript.
  • IGC is a sensitive method to determine phase transition temperatures like polymers' glass transition temperature (TG). On the other hand, TG can be derived through the temperature-volume diagram obtained from MD. It seems beneficial to also study this particular property for further verification when the relevant experiment is already conducted. At least, this can be pointed out in the manuscript as a suggestion for future studies.
  • In Section 3.2 Authors should provide reasonable statements why the solubility parameters related to the long-chain polymer in MD simulation is considered for comparison with the experiment.
  • Authors should severely recheck the whole manuscript for editorial and grammatical mistakes;

Some examples are:

-In line 59, “it” should not refer to the authors.

-In “in acetone” is repeated in line 91.

-in line 34, it should be “non-toxicity and degradablity”

-there is no logical coherence between some successive sentences like in line 49 and some sentences are poorly composed like 61-62, 56- 58

-In line 182, The values of ΔHls, ΔHl∞, and ΔHv of PVA 2488 and PVA 2499 were obtained from table 2 “. They are “presented” in the Table” not “obtained from the table”

-A consistent notation should be assured for PVA polymers (167-171, 199, 217 differs)

- The solubility parameter of PVA, ?_2, should be defined in line 203, the first time it appears in the text.

-in line 247 and 250, “a” and “b” are redundant

-line 312, formula “15” should be “12”.

-line 341, “chapter” should be “paper”.

-line 346, “twoThe inter rater”?

Minor Remarks:

  • In line 118, “force module” should be “Forcite module”
  • (7) is missing in the manuscript for Flory Huggins interaction parameters.
  • The caption of Figure 3, is misleading, “at 298.15 K by extrapolation” is redundant as ?_2 is presented for different temperatures not 298.15 K.
  • The Pressure of NPT ensemble is not defined in section 2.3.2
  • in line 126, “to obtain the most energy” is odd as one should minimize the energy at the beginning of simulation using minimization techniques
  • The acronyms should be defined the first time they appear in the text, as for IGC and PVA in the abstract.
  • The definition for \delta G^sp, i.e. the surface specific adsorption free energy, is better to be used in the abstract rather than the symbol.
  • The way of introducing raw material in 2.1 is not appropriate. It is recommended to use a table or at least some sentences for coherence.

Author Response

Dear Reviewer:

         Thank you for your comments concerning our manuscript entitled “Quantitative study on solubility parameters and surface properties of PVA with different alcoholysis degrees” (ID: polymers-1402432).

Those comments are all valuable and very helpful for revising and improving our paper, as well as the important guiding significance to our researches. We have studied comments carefully and have made correction which we hope meet with approval. Revised portion are marked in blue in the paper. The main corrections in the paper and the responds to the your comments are as flowing:

  1. Response to comment: “The temperature at which the experimental solubility parameter is compared with atomistic simulation is obtained through extrapolation rather than direct measurements while this is the main value for verification of the method. Authors should provide explanation if it relates to any limitation in the experiment.”

It is really true as you suggested that solubility parameter should be obtained through extrapolation. Since the IGC method needs to vaporize the solvent, the solubility parameter at 298.15K cannot be measured, so extrapolation is required. By measuring the solubility parameters at 383.15K, 393.15K, 403.15K, 413.15K, and 423.15K, using the origin software for linear fitting, the linear relationship between the solubility parameters of PVA2488 and PVA2499 and the temperature was obtained, and the solubility parameter at 298.15K was obtained. And we have added this explanation below Table 5.

  1. Response to comment: “For relaxation of amorphous polymer unit cells (besides the single polymer chain) in MD simulation, it is recommended to perform annealing above the glass transition temperature where the higher mobility of polymer chains makes it easier to achieve equilibration within reasonable computational time. For a complete relaxation procedure of polymer unit cells, the Authors are referred to “Izadi, R., Ghavanloo, E., & Nayebi, A. (2019). Elastic properties of polymer composites reinforced with C60 fullerene and carbon onion: Molecular dynamics simulation. Physica B: Condensed Matter, 574, 311636.” and “Malagù, M., Goudarzi, M., Lyulin, A., Benvenuti, E., & Simone, A. (2017). Diameter-dependent elastic properties of carbon nanotube-polymer composites: Emergence of size effects from atomistic-scale simulations. Composites Part B: Engineering, 131, 260-281.””

Thank you very much for your suggestions. Annealing above the glass transition temperature will indeed make it easier to achieve equilibrium within reasonable calculation time. When we designed the experiment, we considered the temperature range of PVA during application, so we determined that the annealing experiment should be carried out in the range of 290K to 590K. The two articles you recommend are very helpful to us, especially in the application of molecular dynamics simulation. In following research, we will anneal above the glass transition temperature as you suggested. We think it will make the experimental results better.

  1. Response to comment: “The Materials Studio software used for the MD simulation should be clearly mentioned and referenced in the manuscript.”

The Materials Studio software used for the MD simulation has been mentioned in the Introduction section. And we have briefly introduced its function and applicable scope.

  1. Response to comment: “IGC is a sensitive method to determine phase transition temperatures like polymers' glass transition temperature (TG). On the other hand, TG can be derived through the temperature-volume diagram obtained from MD. It seems beneficial to also study this particular property for further verification when the relevant experiment is already conducted. At least, this can be pointed out in the manuscript as a suggestion for future studies.”

It is really true as you suggested that the glass transition temperature (TG) can be be derived through the temperature-volume diagram obtained from MD and some researchers are already studying about that. We have pointed out that in the Conclusion section and we will continue to study the molecular dynamics and make more breakthroughs.

  1. Response to comment: “In Section 3.2 Authors should provide reasonable statements why the solubility parameters related to the long-chain polymer in MD simulation is considered for comparison with the experiment.”

We are very sorry for our negligence of the explanation for why the solubility parameters in MD simulation is considered for comparison with the experiment. By comparing the results of the two methods, it can be determined that both methods can be used for parameter measurement. For some parameters that are difficult to be measured by experiment, we can consider using computer simulation to measure. It is conducive to parameter measurement  in this way.

  1. Response to comment: “Authors should severely recheck the whole manuscript for editorial and grammatical mistakes”

We are very sorry for our incorrect writing. All the editorial and grammatical mistakes have been corrected and we have checked the manuscript to make sure there is no such mistake.

We tried our best to improve the manuscript and made some changes in the manuscript. These changes will not influence the content and framework of the paper. We appreciate for your warm work earnestly, and hope that the correction will meet with approval.

Once again, thank you very much for your comments and suggestions.

Reviewer 2 Report

Authors of the manuscript present here the aim of this study - investigationy of the relationship between interfacial properties and structure of PVA matrix composites, and selection of appropriate PVA model for research and application of related composites. The investigations will be interesting in polymer field and the paper could be considered for publication after revision:

*The used in abstract abbreviations should be firstly explained in this part. It would be even better to avoid the abbreviations in the abstract.

*Introduction of the manuscript is rather long. I would recommend that the authors would concentrate only with the description- what is done already with PVA in this research field.

* PVA is very old polymer. The properties of PVA are also very widely investigated. The authors should exactly explain in the paper- what is new developed in their research ?

* What is chemical and physical difference between the samples PVA2488 and PVA2499?

*91: PVA2488 was dissolved in acetone in acetone...

*346: … that twoThe inter rater....

* 357: It is described „This paper, through the study of PVA's surface properties with solubility parameters, as well as interaction parameters, and so on, can provide a valuable reference for the relationship between its structure and properties of composites, so as to make the corresponding composites have better interfacial bonding properties.“   The “relationship between structure and properties” should be declared in the conclusions.  The “composites having better interfacial bonding properties” should be also described in the conclusions.

Author Response

Dear Reviewer:

         Thank you for your comments concerning our manuscript entitled “Quantitative study on solubility parameters and surface properties of PVA with different alcoholysis degrees” (ID: polymers-1402432).

Those comments are all valuable and very helpful for revising and improving our paper, as well as the important guiding significance to our researches. We have studied comments carefully and have made correction which we hope meet with approval. Revised portion are marked in blue in the paper. The main corrections in the paper and the responds to the your comments are as flowing:

  1. Response to comment: “The used in abstract abbreviations should be firstly explained in this part. It would be even better to avoid the abbreviations in the abstract.”

We are very sorry for our incorrect writing and we have explained all abbreviations in Abstract section such as IGC and PVA.

  1. Response to comment: “Introduction of the manuscript is rather long. I would recommend that the authors would concentrate only with the description- what is done already with PVA in this research field.”

It is really true as you suggested that the Introduction should concentrate on its research field and we have revised the introduction part. We have introduced the average degree of polymerization and alcoholysis of PVA and the related research progress. Then we mentioned the interface compatibility of PVA composites, and introduced the research progress. Finally, we explained the main work of this paper.

  1. Response to comment: “PVA is very old polymer. The properties of PVA are also very widely investigated. The authors should exactly explain in the paper- what is new developed in their research”

We have revised the Introduction section and added several researchers and introduced their work. For example, Hailong Xu et al. prepared the porous f-Ti2CTx/PVA foam by the freeze-drying process and it possesses excellent EM absorption ability, and Tatiana V et al. prepared composite films containing poly(vinyl alcohol) with different amounts of graphene oxide and modified the composite films with glycerin.

  1. Response to comment: “What is chemical and physical difference between the samples PVA2488 and PVA2499?”

The difference of PVA2488 and PVA2499 is that they have different alcoholysis degree. We have changed 2.1.1 section and explained the difference between PVA2488 and PVA2499.

  1. Response to comment: “PVA2488 was dissolved in acetone in acetone...” and “… that twoThe inter rater....”

We are very sorry for our incorrect writing. All the editorial and grammatical mistakes have been corrected and we have checked the manuscript to make sure there is no such mistake.

  1. Response to comment: “It is described „This paper, through the study of PVA's surface properties with solubility parameters, as well as interaction parameters, and so on, can provide a valuable reference for the relationship between its structure and properties of composites, so as to make the corresponding composites have better interfacial bonding properties.“ The “relationship between structure and properties” should be declared in the conclusions.  The “composites having better interfacial bonding properties” should be also described in the conclusions.”

We have made correction according to your comments about the Conclusions section. The Conclusions section has been rewritten and extended, and we have introduced that the work done in this paper can make some composite materials have better performance, and propose future research work

We tried our best to improve the manuscript and made some changes in the manuscript. These changes will not influence the content and framework of the paper. We appreciate for your warm work earnestly, and hope that the correction will meet with approval.

Once again, thank you very much for your comments and suggestions.

Reviewer 3 Report

The article: “Quantitative study on solubility parameters and surface properties of PVA with different alcoholysis degrees” describes a study on the determination of Florry-Huggins parameters for two PVA types and various solvents.

The article needs a major revision.

Specific comments:

  1. The title informs that surface properties will be studied – it is misleading because the Florry-Huggins parameter describes phase behavior in polymer blends and solutions. Surface properties would rather be valuable for the case of solid particles dispersed within a polymer matrix.
  2. In many places, the authors suggest that the results can be valuable for PVA composites. Assuming that the authors mean composite of PVA with the dispersed particles it could make sense. However, the text rather suggests that the authors mean polymer blends of thermodynamically incompatible polymer blends.
  3. The introduction is poor. It should be rewritten and extended by adding some examples of PVA composites/blends should be provided. Now it is limited to one, specific example – PLL/PVA. The sense of the creation of PVA composites should be shown.
  4. Introduction – the significance of the study is lacking. The introduction should explain why PVA was selected for research. Who can be potentially interested in the results? Does the water solubility of PVA limit the use of PVA blends?
  5. The full name of IGC should be at the place of its first appearance.
  6. The full name of PLL is lacking.
  7. Discussion and Conclusion sections – the results achieved for the PVA-solvent couples should be referred to polymeric blends (composites).
  8. The manuscript is poorly written. It contains a lot of grammar and editing errors (word repetitions, useless or lacking punctuation, missing letters in words, lack of capital letters at the beginning of the sentence, pva written with small letters).

Author Response

Dear Reviewer:

         Thank you for your comments concerning our manuscript entitled “Quantitative study on solubility parameters and surface properties of PVA with different alcoholysis degrees” (ID: polymers-1402432).

Those comments are all valuable and very helpful for revising and improving our paper, as well as the important guiding significance to our researches. We have studied comments carefully and have made correction which we hope meet with approval. Revised portion are marked in blue in the paper. The main corrections in the paper and the responds to the your comments are as flowing:

  1. Response to comment: “1.The title informs that surface properties will be studied – it is misleading because the Florry-Huggins parameter describes phase behavior in polymer blends and solutions. Surface properties would rather be valuable for the case of solid particles dispersed within a polymer matrix.”

It is really true as you suggested that the title is misleading. The parameters measured in this article, such as solubility parameters, belong to the surface properties of the material, and our purpose is to quantitatively study the surface properties of PVA with different alcoholysis degrees. After our cautious consideration, we changed the title to “Quantitative study on solubility parameters and related ther-modynamic parameters of PVA with different alcoholysis degrees”.

  1. Response to comment: “2.In many places, the authors suggest that the results can be valuable for PVA composites. Assuming that the authors mean composite of PVA with the dispersed particles it could make sense. However, the text rather suggests that the authors mean polymer blends of thermodynamically incompatible polymer blends.”

Thank you very much for your suggestions. We are very sorry for that we did not explain clearly. The main purpose of this research is to explore the surface properties of PVA and some natural polymers such as proanthocyanidins and lignin, and to make a reasonable explanation for the interaction, so as to provide reference for the further research on the composite materials of PVA and natural polymer.

  1. Response to comment: “3.The introduction is poor. It should be rewritten and extended by adding some examples of PVA composites/blends should be provided. Now it is limited to one, specific example – PLL/PVA. The sense of the creation of PVA composites should be shown.”

We have revised the Introduction section and added several researchers and introduced their work. For example, Hailong Xu et al. prepared the porous f-Ti2CTx/PVA foam by the freeze-drying process and it possesses excellent EM absorption ability, and Tatiana V et al. prepared composite films containing poly(vinyl alcohol) with different amounts of graphene oxide and modified the composite films with glycerin.

  1. Response to comment: “4.Introduction–the significance of the study is lacking. The introduction should explain why PVA was selected for research. Who can be potentially interested in the results? Does the water solubility of PVA limit the use of PVA blends?”

We are very sorry for our negligence of the significance of this study and other information. We have revised the Introduction section and Conclusions section and explained the significance of the study. At the same time, we also pointed out the future research direction in Conclusions section.

  1. Response to comment: “5.The full name of IGC should be at the place of its first appearance.” and “6.The full name of PLL is lacking.”

We are very sorry for our incorrect writing and we have added the full name of IGC. And we have explained all abbreviations such as IGC and PVA .

  1. Response to comment: “7. Discussion and Conclusion sections – the results achieved for the PVA-solvent couples should be referred to polymeric blends (composites).”

We have made correction according to your comments. This study can provide a reference for improving the performance of PVA composite materials such as PVA hydrogel and PVA composite film. And the PVA molecular dynamics method can also be used in other studies to verify or directly calculate the corresponding parameters, which is very helpful for some data difficult to measure.

  1. Response to comment: “8. The manuscript is poorly written. It contains a lot of grammar and editing errors (word repetitions, useless or lacking punctuation, missing letters in words, lack of capital letters at the beginning of the sentence, pva written with small letters).”

We are very sorry for our incorrect writing. All the editorial and grammatical mistakes have been corrected and we have checked the manuscript to make sure there is no such mistake.

We tried our best to improve the manuscript and made some changes in the manuscript. These changes will not influence the content and framework of the paper. We appreciate for your warm work earnestly, and hope that the correction will meet with approval.

Once again, thank you very much for your comments and suggestions.

Round 2

Reviewer 2 Report

After the revision I would like to recommend for Editor consider acceptance of the manuscript for Polymers.

Author Response

Dear Reviewer:

    Thank you for your comments concerning our manuscript entitled “Quantitative study on solubility parameters and surface properties of PVA with different alcoholysis degrees” (ID: polymers-1402432).

    Those comments are all valuable and very helpful for revising and improving our paper, as well as the important guiding significance to our researches. We have studied comments carefully and have made correction which we hope meet with approval. Revised portion are marked in blue in the paper.

    The introduction part has been revised; we have introduced the average degree of polymerization and alcoholysis of PVA and the related research progress. Then we mentioned the interface compatibility of PVA composites and introduced the research progress. We also added several researchers and introduced their work. Finally, we explained the main work of this paper.

    The Conclusions section has been rewritten and extended, and we have introduced that the work done in this paper can make some composite materials have better performance, and propose future research work.

    We thank you very much for pointing out some minor issues in our manuscript, for example, we should explain all abbreviations in Abstract section such as IGC and PVA, and we need explain the difference between PVA2488 and PVA2499. And all the editorial and grammatical mistakes have been corrected and we have checked the manuscript to make sure there is no such mistake.

    We have revised the manuscript according to your comments last time. We appreciate for your warm work earnestly, and hope that the correction will meet with approval.

    Once again, thank you very much for your comments and suggestions.